# Trading Monotonicity for Cost in Beam Search

**Sofia Lemons,**[1,2] **Carlos Linares López,**[3] **Robert C. Holte,**[4] **Wheeler Ruml**[1]

[1] University of New Hampshire
[2] Earlham College
[3] Computer Science and Engineering Department, Universidad Carlos III de Madrid
[4] University of Alberta, Alberta Machine Intelligence Institute (Amii)
sofia.lemons@earlham.edu, carlos.linares@uc3m.es, rholte@ualberta.ca, ruml@cs.unh.edu

## Abstract

Beam search is a popular satisficing heuristic search algorithm, but increasing the beam width sometimes causes the algorithm to return a worse solution. A recent variant of beam search, monobead, guarantees nonincreasing solution cost with increasing beam width. However, the monotonicity of monobead sometimes comes at the price of increased cost and time for small beam widths. In this paper, we explore two algorithmic variants that lie between beam search and monobead. We find that, as hoped, our hybrids of beam and monobead can often find solutions with better cost than monobead and typically have more monotonic behavior than bead. This work improves our understanding of the price of monotonicity in beam search.

## Introduction

Beam search (Bisiani 1987) is a popular satisfying heuristic search algorithm that can be used to solve problems that are too large to solve with more exhaustive best-first search algorithms. It proceeds as breadth-first search, except that at every depth level of the search, given a width parameter $w$, beam search selects the $w$ best children of the nodes at the previous level and discards all the rest. This leads to an algorithm that is neither complete nor optimal, but can often find solutions to problems that would be infeasible to solve with more exhaustive search methods.

However, beam search also suffers from the problem that providing a wider beam may lead the algorithm to discard nodes that were explored at smaller beam widths and return a solution that is worse than may have been found with a smaller beam. This behavior is demonstrated in Figure 1 (drawn from (Lemons et al. 2022)), where we see beam search providing very inconsistent solution cost on a 15-puzzle instance as the beam width varies.

This non-monotonic behavior can be caused by *cuckoo nodes*, (Lemons et al. 2022) which are nodes whose heuristic values are incorrectly low. They will be selected for the beam over nodes which may lead to better solutions, and since there is limited space on the beam, those nodes leading to better solutions may be lost from the search altogether.

Cohen and Beck (2019) studied the degradation of solution quality in beam search in the specific domain of neural sequence models and address it in a domain-specific manner. Vadlamudi, Aine, and Chakrabarti (2013) proposed an

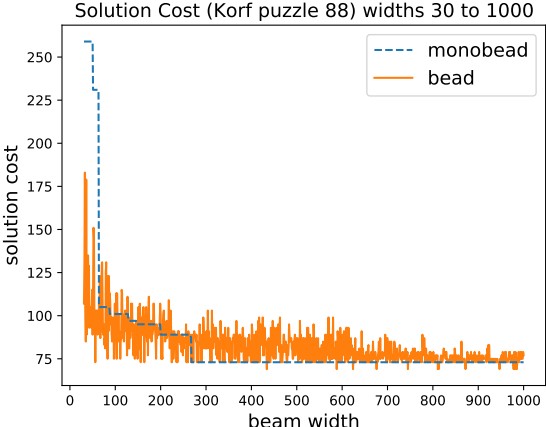

Figure 1: Solution cost as beam width varies (unit cost.)

algorithm, Incremental Beam Search, which attempts to return monotonically improving solutions in an anytime fashion, but which relies on an additional parameter of a maximum depth and which is not monotonic across multiple executions of the algorithm with different maximum widths or maximum depths.

Lemons et al. (2022) introduce the *monobeam* algorithm, which performs a beam-style search but in a way guaranteed to return solutions with monotonically improving solution quality as beam width increase. The set of nodes at each level is considered to be ordered; each indexed position is called a beam slot. The algorithm expands from and fills the beam slots one at a time, preventing the nodes found in higher beam slots from affecting the search order in lower beam slots. Because of this, the algorithm is guaranteed, for any beam width $w$, to return a solution with cost less than or equal to what would be returned by monobeam with any width less than $w$. In Figure 1, we see this behavior as the algorithm finds a solution at a given beam width, then retains that quality of solution for later beam widths until a better solution is found.

It has been shown in other settings (Thayer and Ruml 2009) that in non-unit cost domains, it can be beneficial for best-first search algorithms to search using distance-to-go estimates ($d$), as opposed to cost-to-go estimates ($h$) in

ordering the search. Lemons et al. (2022) show that beam search can also benefit significantly from ordering the search on $l(n) = depth(n) + d(n)$ instead of $f(n)$. They introduce the bead and monobead algorithms, which perform significantly better than beam and monobeam in non-unit cost domains (and are equivalent to them in unit cost domains). Thus we focus on these algorithms in this paper.

The guarantee of monotonicity, while making beam search easier to use, often comes with a price in terms of solution cost. We can see in Figure 1 many points at which bead search returns a better quality solution than monobead, and this is true across the wide range of experiments reported by Lemons et al. (2022). In this paper, we explore whether it is possible to attain some of the benefits of monotonicity while reducing the penalty paid in terms of solution cost. An obvious approach is to select only some elements of the beam using a monotonic selection policy. We present two new algorithms that each embody a different way of doing this. First, mono-floor selects the first portion of the beam (beam slots 1 through $w-n$, where $w$ is beam width) monotonically but then selects the remainder of the beam using the regular bead selection rule. And second, mono-onward selects the first $n$ slots of the beam as in bead but then the remaining slots are selected using monotonic slot-based selection. We study the properties and the empirical performance of these variants on four popular search benchmarks, using a variety of cost models in two of those domains. We find that, as hoped, mono-onward and mono-floor often find solutions with better cost than monobead and typically have more monotonic behavior than bead. This work helps illuminate the design space between regular and monotonic beam searches, illustrating what is possible, providing practitioners with additional tools for satisfying search problems, and clarifying possible tradeoffs between monotonicity and solution cost.

## Background

The reason why beam search often shows non-monotonic behavior is because when the width is increased, new nodes are encountered and expanded, and the children of those new nodes may have incorrectly low cost-to-go estimates. The children of these new nodes look more promising than others generated lower in the beam and take a position in the next beam, pushing out the children of the other nodes. These nodes are termed *cuckoo nodes*. If such cuckoo nodes are selected for the beam, the search may lose access to good quality solutions that were found by beam searches with lower widths.

Monobeam is a beam search algorithm that, unlike regular beam search, provides guarantees on monotonicity, i.e., the solution found with a beam width $w_2$ is never worse than any solution found with a beam width $w_1 < w_2$. Its main difference with regular beam search is that after expanding the $i$-th node of the current beam, it selects the best child among all children of the first $i$ expanded nodes, to be in the next beam. This selection procedure guarantees that the nodes picked for the next beam resulted from parents in the corresponding beam slot or lower, preventing the appearance of cuckoo nodes and thus guaranteeing monotonicity. Proof

---

**Algorithm 1:** monobeam(start,width)

1   solutionCost $\leftarrow \infty$;
2   beam[1] $\leftarrow$ start;
3   **while** *at least one slot in the beam has a node with $f$ value < solutionCost* **do**
4     candidates $\leftarrow \emptyset$, nextBeam $\leftarrow$ [];
5     **for** *each beam slot c from 1 to width* **do**
6       **if** *beam[c] is a node* **then**
7         **for** *each child of beam[c]* **do**
8           **if** *f(child) < f(beam[c])* **then**
9             f(child) $\leftarrow$ f(beam[c]);
10           **if** *child is a goal and f(child) < solutionCost* **then**
11             store as solution;
12             solutionCost $\leftarrow$ f(child);
13           **else**
14             add child to candidates;
15       **if** *candidates is nonempty* **then**
16         nextBeam[c] $\leftarrow$ remove min $f$-value node from candidates;
17     beam $\leftarrow$ nextBeam;
18   return solution;

---

of this property of monobeam was given by Lemons et al. (2022). Algorithm 1 shows the main core of monobeam, where duplicate detection and solution cost pruning have been intentionally left out to improve readability.

Duplicate detection and solution cost pruning need to be handled with care to preserve monotonicity. Removing a node from further consideration because it was previously encountered with a better $f$ value might introduce non-monotonic behavior if the node previously seen were generated in a higher beam slot. Neither beam search nor monobeam are complete algorithms and thus, there is no guarantee that we will fully explore the paths available from the first time we encountered the node. Removing the duplicate may cause us to fail to find a solution that can be reached through it. Consequently, when a good solution can be found with a beam width low enough to prevent the generation of the node at the shallower level, incrementing the beam width might prevent finding the same solution. However, if the node previously seen at the shallower level was generated in a lower beam slot, then the duplicate can be safely removed because doing so does not remove any solution found through the node at the shallower level.

When a solution is found in non-unit cost domains, there is still a possibility that a better solution may have been available at a greater depth in the search space. Standard beam search implementations can exploit or ignore this fact without loss of their important features. However, monobeam must continue searching until it is certain there are no better solutions available, because one of those deeper solutions may have been found by a lower width search. Monobeam can, however, use the cost of the incumbent solution to prune away nodes with an $f$ value greater than that incumbent's cost, so long as the heuristic is admissible and

**Algorithm 2:** mono-floor(start,width,n)

```
1  solutionCost ← ∞;
2  beam[1] ← start;
3  while at least one slot in the beam has a node with l
      value < solutionCost do
4  │   candidates ← ∅, nextBeam ← [];
5  │   for each beam slot c from 1 to width do
6  │   │   if beam[c] is a node then
7  │   │   │   for each child of beam[c] do
8  │   │   │   │   if l(child) < l(beam[c]) then
9  │   │   │   │   │   l(child) ← l(beam[c]);
10 │   │   │   │   if child is a goal and l(child) <
      │   │   │   │     solutionCost then
11 │   │   │   │   │   store as solution;
12 │   │   │   │   │   solutionCost ← l(child);
13 │   │   │   │   else
14 │   │   │   │   │   add child to candidates;
15 │   │   if c ≤ width − n and candidates is nonempty
      │   │     then
16 │   │   │   nextBeam[c] ← remove min l-value node
      │   │     from candidates;
17 │   for each beam slot c from width − n + 1 to width
      │     do
18 │   │   if candidates is nonempty then
19 │   │   │   nextBeam[c] ← remove min l-value node
      │   │     from candidates;
20 │   beam ← nextBeam;
21 return solution;
```

the algorithm is selecting nodes based on $f$ values. This is because no node will be pruned away which could lead to a better solution, and we will still select for the same slots any node with $f$ value lower than the incumbent cost. However, when searching on $l$ values (recall $l(n) = depth(n) + d(n)$), we cannot use the incumbent's cost for pruning. We would have to prune on $f$ value to insure that we give the best solution (not just the shallowest, which $l$ optimizes), but our node selection would be disrupted because we may lose some of the nodes which would have been selected by a lower width search based on having the best $l$ value.

Because of the demonstrated performance of distance-to-go in this setting, we focus our discussion and experiments henceforth on the $d$-based variants of beam, monobeam, and our new algorithms. In unit cost settings, the $d$-based and $h$-based variants are equivalent, but we will use the names with 'd' in them (bead, monobead, etc.) throughout.

Unfortunately, monobead can sometimes produce worse solution costs than bead search for a specific beam width. Monobead is forced to choose from a limited subset of the nodes when filling a specific slot, rather than being able to consider all of the nodes. While this prevents the appearance of cuckoo nodes, it also sometimes prevents the selection of nodes that are more promising.

## Mono-Floor

We first present a hybrid approach, called mono-floor, which merges the slot-based monotonic approach with a non-monotonic beam-style search. (Pseudocode is presented in Algorithm 2.) It takes two arguments: the beam width $w$ and a non-monotonicity parameter $n$. For a beam width $w$, beam slots 1 through $w − n$ are selected monotonically, that is, the node from a given slot is expanded, its children added to a priority queue called *candidates*, and the corresponding slot in the next beam is filled using the best node in *candidates* at that time. This allows us to ensure that, in this portion of the beam, we maintain search order through each slot $s$ as if a search of width $s$ were being performed, for all $s ≤ w − n$. In this way, we avoid the influence of cuckoo nodes and ensure that we will not lose any solution that would have been found by a search with a narrower beam width. However, once we reach the upper $n$ slots of the beam, we proceed to expand all nodes from slots $(w − n) + 1$ onward, adding their children to *candidates* and then select for slots $(w − n) + 1$ through $w$ the minimum $l$−value nodes now on *candidates*. This portion of the beam acts more like a standard beam search, optimizing for $l$ value and taking no measures to preserve solutions that would have been found by a search with a narrower beam width.

The careful preservation of search order in the first portion of the beam means that a mono-floor search of width $w$ will return no worse of a solution than could have been found by a monobead search of width $w − n$. It may find better solutions than this bound in the non-monotonic portion of the beam, but there is no certainty that these will be maintained when the beam width changes. While this is not true monotonic behavior across beam widths, it allows us to have some kind of fall-back guarantee of solution quality when the upper portion of the beam is ill-behaved. This floor on solution quality is what gives rise to the name mono-floor.

Figure 2 shows the behavior of mono-floor where $n = 30$ relative to bead search on a single instance of the 15-puzzle. It is clear that mono-floor is not strictly monotonic. However, there are many sequences of beam widths where the solution quality is stable, which beam search does not have in this instance. When a solution is found in the lower portion of the beam, that solution is maintained in searches with larger beam widths. However, the solution cost does lower at some single beam widths without maintaining that solution quality for subsequent beam widths. These are cases where a new solution was found in the upper, non-monotonic portion of the beam but then lost at higher beam widths.

Similar to monobeam and monobead, duplicate elimination can be done without loss of monotonicity if it is done with attention to the slot in which a duplicate was expanded. If we only eliminate duplicates from slots greater than or equal to the slot at which the node was previously seen, this will still guarantee that search order in lower slots will not be affected by search done in higher beam slots. The implementations tested in unit cost domains used both duplicate elimination and solution cost pruning, while the implementations using distance-to-go in non-unit cost domains used only duplicate elimination but not pruning based on solution cost.

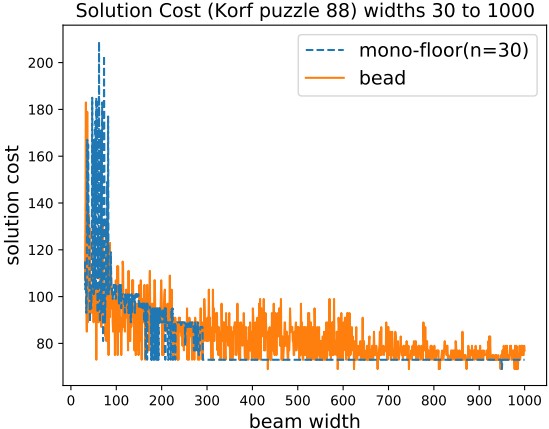

Figure 2: Solution cost as beam width varies (unit cost.)

Also like monobead, in non-unit-cost domains mono-floor cannot terminate immediately upon finding a solution, as it might find a shallower solution with greater cost than a mono-floor search with a narrower beam width. It can terminate only when there are no nodes remaining that have an $f$ value lower than the cost of the current incumbent solution. An f-based search could prune nodes with $f$ values greater than or equal to the cost of the incumbent solution. The pruning would still be guaranteed not to remove a solution that would be found at a lower beam width. However, when using distance-to-go, we cannot prune in this way.

## Experimental Results

We implemented all algorithms in C++ [1] and tested their behavior on several classic search benchmarks. Each algorithm was run with beam widths 30, 100, 300, 1000, 3000, 10000, 30000, and 100000 for results shown in Figures 4 and 5, and all widths 1 through 1000 for results in Table 1. Algorithms were given a memory limit of 7.5GB. We tested mono-floor with n values of 30, 100, 300, and $\frac{w}{2}$, where beam width is $w$.

The sliding tile puzzle experiments use five cost models: unit cost, where the cost of moving any tile is 1; heavy cost, moving tile numbered $t$ costs $t$; sqrt cost, moving tile $t$ costs $\sqrt{t}$; inverse cost, $1/t$; reverse cost, moving tile t costs $16 - t$. The cost-to-go heuristic was a weighted version of the Manhattan distance in which each tile's Manhattan distance is multiplied by the cost of moving that tile. Our implementation expands nodes at a rate of approximately 1.5 million nodes per second. The standard Korf (1985) 100 15-puzzles were used in all cost models.

For testing on the pancake problem, 50 random instances of pancake problems with stacks of 20, 50, and 70 pancakes were used. The gap heuristic (Helmert 2010) was used by all algorithms.

In the blocks world domain, we tested on 100 random instances with 20 blocks total, with two different action models: one in which blocks are directly moved to a stack as

an action ('blocks world') and one in which picking up and putting down blocks each use an action, so therefore the branching factor is smaller and plans are longer ('deep blocks world').

Across nearly all domains tested (displayed in Figure 4), mono-floor with all values of n provided solutions with average costs between those provided by bead and monobead. This includes finding solutions better than bead in the heavy cost 20-pancake problem, and some mono-onward configurations giving the best solutions overall at a few beam widths in 50-pancake and 70-pancake problems. Given that the only guarantee on mono-floor's solution quality is that it will return solutions at least as good as monobead run with width $w - n$, this demonstrates that a significant benefit in solution quality is being provided by the non-monotonic portion of the beam.

Solutions provided from the non-monotonic portion of the beam can reduce the monotonicity of mono-floor at least some of the time, as seen in Figure 2. To quantitatively measure the monotonicity of each algorithm, we ran the algorithms using beam widths of 1 through 1000 and then computed Kendall's $\tau$ rank correlation statistic between the beam width used and the solution cost returned (Kendall 1955). This statistic compares the solution costs at all pairs of beam widths and computes the probability that a pair is in non-increasing order.[1] We then calculated the mean of these measurements across all instances tested in that domain. Table 1 presents the results of this analysis. For bead search in most domains, there is significant non-monotonic behavior. The general trend of increasing solution quality leads to a generally positive correlation, and in some settings it is very high, such as unit and heavy pancake and heavy and square root cost tiles. Across all domains, mono-floor with $n = \frac{w}{2}$ provided as good a rank correlation as bead or higher. Mono-floor with $n = 30$ consistently provided near-perfect rank correlation, indicating a very high amount of monotonicity. When $n = 100$, mono-floor often provided significantly higher monotonicity than bead, but occasionally was equal to bead or slightly lower, such as in heavy and square root cost tiles and the unit 20-pancake problem. Mono-floor with $n = 300$ is even less consistent and in several settings has a lower rank correlation than any of the algorithms, showing that there is a limit to how far we can safely extend the non-monotonic portion of the beam.

## Mono-Onward

It is likely that a practitioner does not need monotonicity throughout the entire range of beam widths. For example, perhaps the minimum beam width that will be considered is 100. There is no reason, then, to pay the price of monotonicity for those first 100 slots. This leads us to mono-onward, in which the first $n$ slots of the beam are selected from all children of the first $n$ parents, as in standard beam search, and then the remainder are expanded and selected one slot

---

[1]Code available at https://github.com/snlemons/search.

[1]That is to say, we compute a variant of $\tau$ in which ties are counted as concordant rather than discordant. Furthermore, we counted the cost of a run that fails to find a solution as the maximum solution cost of any width + 1.

| | bead | mono-floor | | | | mono-onward | | | |
|---|---|---|---|---|---|---|---|---|---|
| | | $n=\frac{w}{2}$ | n=30 | n=100 | n=300 | $n=\frac{w}{2}$ | n=30 | n=100 | n=300 |
| tiles (unit) | 0.76 | 0.83 | 0.98 | 0.90 | 0.78 | 0.77 | 1.00 | 0.98 | 0.93 |
| tiles (heavy) | 0.93 | 0.95 | 0.98 | 0.93 | 0.91 | 0.95 | 1.00 | 0.99 | 0.98 |
| tiles (sqrt) | 0.95 | 0.95 | 0.99 | 0.94 | 0.91 | 0.95 | 1.00 | 0.99 | 0.98 |
| tiles (reverse) | 0.88 | 0.95 | 0.99 | 0.96 | 0.89 | 0.91 | 1.00 | 0.99 | 0.97 |
| tiles (inverse) | 0.84 | 0.94 | 0.99 | 0.95 | 0.89 | 0.91 | 1.00 | 0.99 | 0.97 |
| 20pancake (unit) | 1.00 | 1.00 | 1.00 | 1.00 | 1.00 | 1.00 | 1.00 | 1.00 | 1.00 |
| 20pancake (heavy) | 0.97 | 1.00 | 1.00 | 1.00 | 1.00 | 1.00 | 1.00 | 1.00 | 1.00 |
| 20bw | 0.84 | 0.88 | 0.96 | 0.90 | 0.82 | 0.80 | 1.00 | 0.98 | 0.93 |
| 20bwdp | 0.77 | 0.87 | 0.97 | 0.90 | 0.82 | 0.75 | 1.00 | 0.98 | 0.88 |

Table 1: Rank correlation measurements across domains.

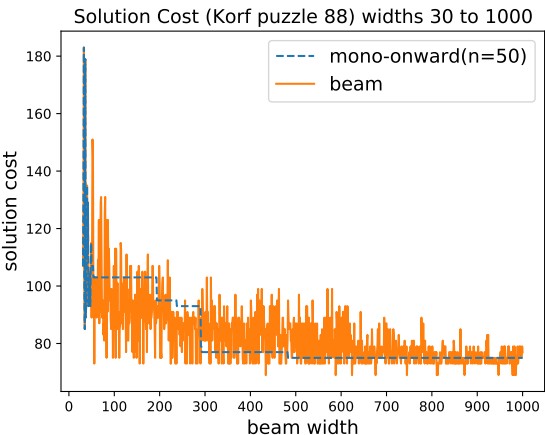

Figure 3: Solution cost as beam width varies (unit cost.)

at a time for the remainder of the beam. (As in mono-floor, $n$ refers to the number of slots selected non-monotonically.) Pseudocode is presented in Algorithm 3. Mono-onward is guaranteed to be monotonic for widths $\geq n$ but, like regular beam search, has no guarantee on performance for widths below $n$. The algorithm can benefit from the effectiveness of standard beam search in the lower beam slots, while being careful not to lose good solutions for beam widths greater than $n$. This could allow a practitioner who knows that a solution can often be found by standard beam search with width $n$, to then assign more resources to the search and know it will improve upon what would be found by beam search with width $n$.

Figure 3 shows the behavior of mono-onward for a fixed $n$ of 50, as compared to standard beam search. We see that for beam widths lower than 50, it is as non-monotonic as regular beam search. However, for beam widths higher than 50 it demonstrates perfectly monotonic behavior. It is noteworthy that there are beam widths lower than 50 where it returns solutions better than the one returned at beam width 50. If the $n$ value were set to one of these, that solution would provide the value for the starting flat portion of the graph, but that value could be different for each problem instance and is not necessarily known in advance.

For duplicate elimination, mono-onward treats all slots 1 through $n$ as being the same slot so duplicates can be eliminated based on a past node found anywhere in this range. For slots $n+1$ through $w$, the usual monobead duplicate elimination rules are applied, where a duplicate can only be eliminated based on a node which was expanded from a slot less than or equal to the current one.

As with monobead and mono-onward, in non-unit cost domains mono-onward cannot terminate after finding the first solution, but must continue until no nodes remain with $f$ value less than the incumbent solution's. Likewise, pruning nodes based on the cost of an incumbent solution could be done without losing monotonicity guarantees if the algorithm was ordering nodes based on $f$ values, but is not used when ordering nodes on $l$ values.

## Experimental Results

We tested mono-onward on all domains listed in the previous section, with n values of 30, 100, 300, and $\frac{w}{2}$. Results are displayed in Figure 5 and Table 1.

Mono-onward with all values of n generally provides solutions with costs between those of bead and monobead across all domains tested, with one exception where it is worst and one exception where it is best. In the unit cost 20-pancake problem (Figure 5f), there are two beam widths where mono-onward with $n = \frac{w}{2}$ provides the worst solution qualities and at one of these beam widths mono-onward with $n = 100$ ties with it for these worst quality solutions. At the lowest beam width (30), mono-onward with $n = \frac{w}{2}$ likely suffers from not having a large enough width devoted to either of its search strategies to provide a good solution. At the third beam width (300), there may be some instances where the non-monotonic portion of the beam uses a width which performs poorly for standard beam search, and the monotonic portion is too constrained by its selection rules to find a better solution. We could see this sort of stall in solution quality due to the n value being at a poorer beam width earlier in Figure 3, where the solution cost flattened out to the solution found at beam width 50 (because $n = 50$ in that execution), and it took significant extra width to reach a solution as good as some of those found at lower beam widths.

Mono-onward with a non-fixed $n$ value provides no guarantees on monotonicity, because the pool of nodes available in the slot-based upper portion of the beam will be impacted by the nodes in the non-monotonic lower portion of the

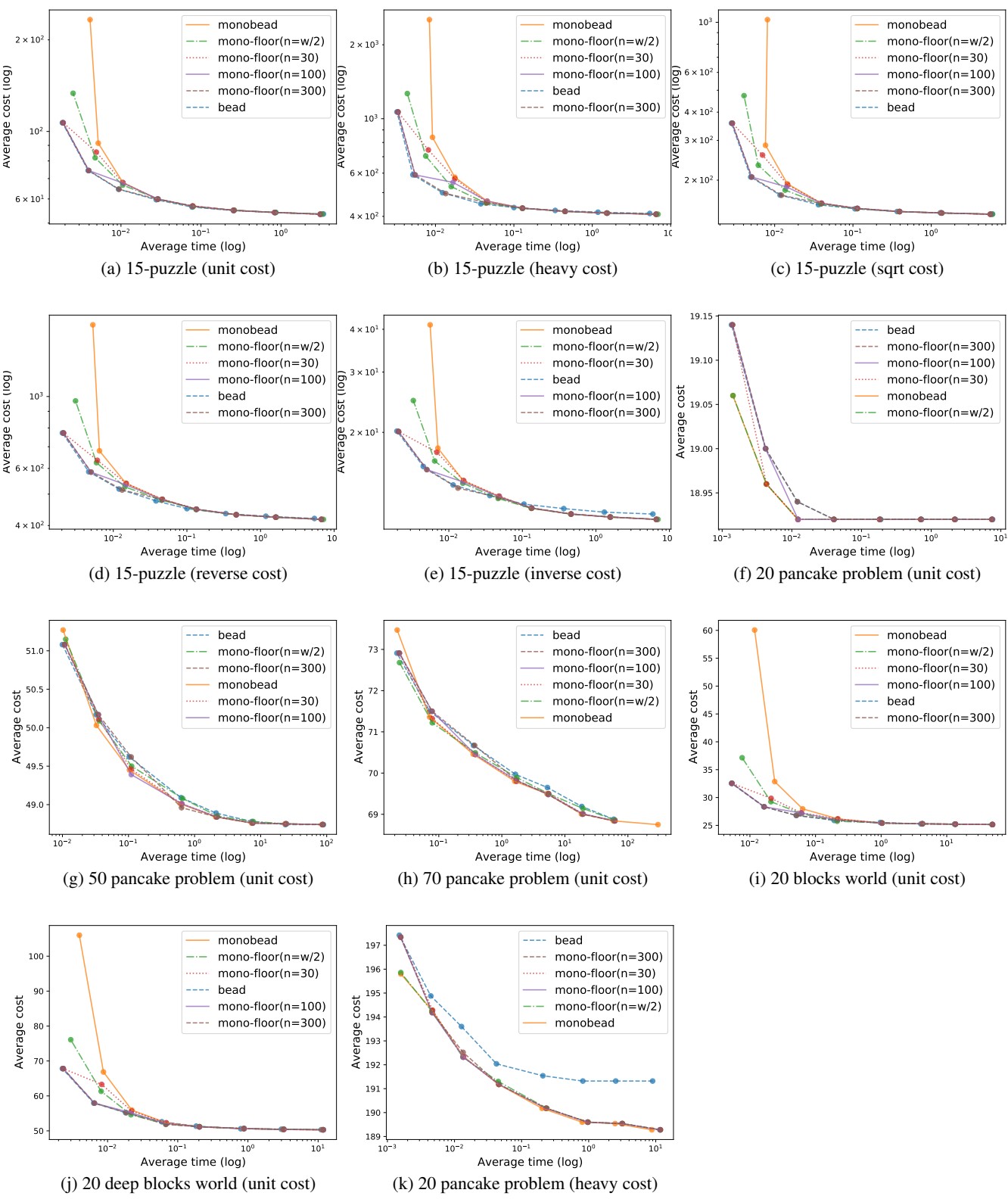

(a) 15-puzzle (unit cost)

(b) 15-puzzle (heavy cost)

(c) 15-puzzle (sqrt cost)

(d) 15-puzzle (reverse cost)

(e) 15-puzzle (inverse cost)

(f) 20 pancake problem (unit cost)

(g) 50 pancake problem (unit cost)

(h) 70 pancake problem (unit cost)

(i) 20 blocks world (unit cost)

(j) 20 deep blocks world (unit cost)

(k) 20 pancake problem (heavy cost)

Figure 4: Time versus cost as beam width is varied.

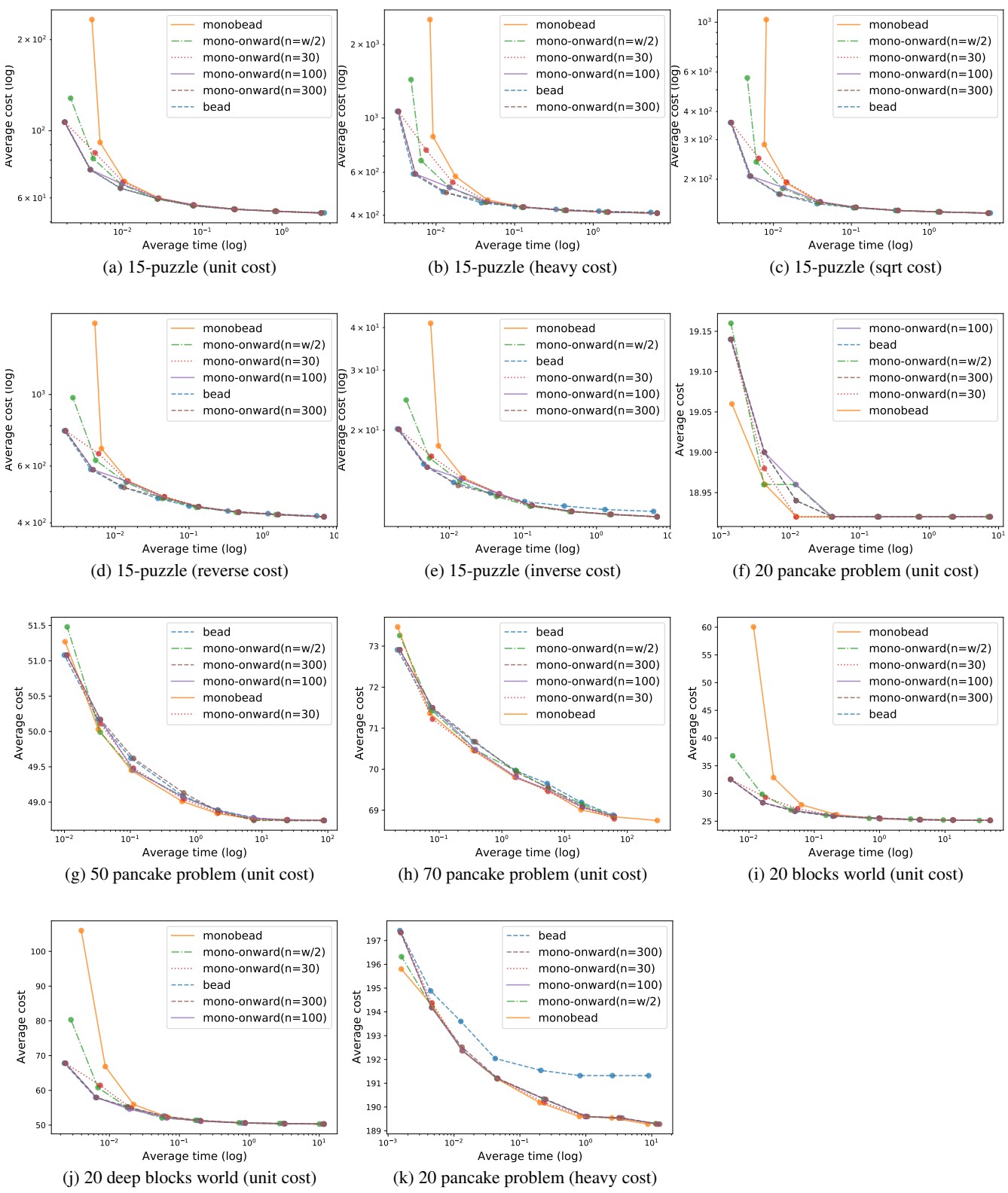

Figure 5: Time versus cost as beam width is varied.

**Algorithm 3:** mono-onward(start,width, n)

```
 1  solutionCost ← ∞;
 2  beam[1] ← start;
 3  while at least one slot in the beam has a node with l
       value < solutionCost do
 4  |   candidates ← ∅, nextBeam ← [];
 5  |   for each beam slot c from 1 to n do
 6  |   |   if beam[c] is a node then
 7  |   |   |   for each child of beam[c] do
 8  |   |   |   |   if l(child) < l(beam[c]) then
 9  |   |   |   |   |   l(child) ← l(beam[c]);
10  |   |   |   |   if child is a goal and l(child) <
       |   |   |   |     solutionCost then
11  |   |   |   |   |   store as solution;
12  |   |   |   |   |   solutionCost ← l(child);
13  |   |   |   |   else
14  |   |   |   |   |   add child to candidates;
15  |   for each beam slot c from 1 to n do
16  |   |   if candidates is nonempty then
17  |   |   |   nextBeam[c] ← remove min l-value node
       |   |   |     from candidates;
18  |   for each beam slot c from n+1 to width do
19  |   |   if beam[c] is a node then
20  |   |   |   for each child of beam[c] do
21  |   |   |   |   if l(child) < l(beam[c]) then
22  |   |   |   |   |   l(child) ← l(beam[c]);
23  |   |   |   |   if child is a goal and l(child) <
       |   |   |   |     solutionCost then
24  |   |   |   |   |   store as solution;
25  |   |   |   |   |   solutionCost ← l(child);
26  |   |   |   |   else
27  |   |   |   |   |   add child to candidates;
28  |   |   if candidates is nonempty then
29  |   |   |   nextBeam[c] ← remove min l-value node
       |   |   |     from candidates;
30  |   beam ← nextBeam;
31  return solution;
```

beam. In spite of this, mono-onward with $n = \frac{w}{2}$ provides greater rank correlation values than bead in the majority of domains tested, as seen in Table 1. Even more noteworthy, mono-onward with fixed $n$ values (30, 100, and 300) provided consistent improvement in rank correlation over bead search in all cases except unit 20-pancake, where all algorithms had a score of 1.00. There is a typically gradual decrease in rank-correlation scores as the $n$ value increases, because of two factors. First, as $n$ increases, more of the executions (all those with $w < n$) will be using a standard beam search approach. Second, the solution found at beam width $n$ may be worse than some solution found earlier. This will cause one or more plateaus in solution quality until the algorithm discovers a new solution in the monotonic portion of the beam. We can see an example of this in Figure 3 for beam widths 50 through around 300.

## Discussion

The amount of the beam dedicated to monotonic search or standard beam search has an impact both on solution quality and monotonicty, but it is not clear in advance what the tradeoff will be for a given value of $n$. Users of these algorithms will likely want to be able to balance these qualities more precisely for their specific setting. There may be useful work to be done around tuning the $n$ parameter or providing guidance on what $n$ value is appropriate for a particular problem.

It is not well known when beam search is more or less inclined toward non-monotonic behavior. Its monotonicity varies across domains, cost models, and even specific instances. Future work should be done to better understand what features of a domain, problem, or beam width lead to non-monotonic behavior.

## Conclusions

Given the lack of scalability of optimal heuristic search, satisficing methods such as bead and monobead are vitally important for enabling applications. We have shown that it is possible to design algorithms that lie between bead and monobead. Both mono-floor and mono-onward provide a parameter that allows the user adjust their degree of monotonicity. Our experimental study showed that this indeed reduces the price of monotonicity in practice, resulting in the hoped-for tradeoff between monotonicity and cost. In the case of mono-floor, we can be sure not to fall back lower than the 'floor' of a solution found by monobead using width $w - n$. And in mono-onwards, we are certain there will be no increase in solution cost for all beam widths greater than $n$. These algorithms serve as additional examples of variants of beam search for which, unlike the original beam search, some kind of behavior guarantee can be made.

## Acknowledgments

We are grateful for support from the NSF-BSF program via NSF grant 2008594.

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
