# OpenReview forum: "Trading Monotonicity for Cost in Beam Search"
_icaps-conference.org/ICAPS/2022/Workshop/HSDIP — HSDIP 2022_

### Official Review · Reviewer_MPCn · 2022-04-21
**Good empirical evaluation, limited list of references**

**Confidence:** 4
**Overall Score:** Weak Accept

**Review:**

In this paper, the authors discuss two additional variants of the known beam search method. The recent variant of beam search, “monobead”, guarantees nonincreasing solution cost with increasing beam width, however, the monotonicity of monobead sometimes increases cost and time for small beam widths. Following this result, the authors explore the trade-off between monotonicity and solution cost and suggest two new algorithms. The first one, mono-floor selects the first portion of the beam monotonically but then selects the remainder of the beam using the regular bead selection rule. And second, mono-onward selects the first n slots of the beam as in bead but then the remaining slots are selected using monotonic slot-based selection. For both algorithms, a vast empirical evaluation is presented.

The paper is well written and relevant to the workshop. It also provides a good overall framework to introduce the problem, however, I would expect to see an example to introduce the “cuckoo nodes” and not just a pointer to the plot where the issue is examined. In addition, I would also expect to have proof to establish some of your claims for example: “This selection procedure guarantees that the nodes picked for the next beam resulted from parents in the corresponding beam slot or lower, preventing the appearance of cuckoo nodes and thus guaranteeing monotonicity.” and others.

The paper structure is somewhat unique, but in this case, it makes sense to discuss and present the empirical evaluation of each of the algorithms separately. The results show some improvement compared to other techniques and it is a contribution. I would like the authors to discuss the difference between the two approaches and why both of them are needed. There is a minor discussion in the Conclusions section but I would expect to give it a more substantial part in the paper and answer which of the two algorithms is better, when to use what, etc.
The Experimental Results sections are mostly good but there are too many plots. It is better to have specific ones that show the point that you would like to extenuate, such as significantly good results or bad ones, and elaborate on the others as part of the discussion. When having so many plots it is hard to focus on the main point you ask the readers to take from the experiment part. There are also minor issues with the presentation itself: Not sure what is the order of the plots – pancake then blocks and pancake again? Legend is not consistent and also the “average cost” is not consistently located.

My main concern is the very limited list of references. You start the paper with “Beam search is a popular satisfying heuristic search algorithm” but suggest only 4 relevant references?

---

> ### Author Response · Authors · 2022-04-29
> **Author response**
>
> Many thanks for your helpful suggestions, which we will take into account in the camera-ready version if the paper is accepted to the workshop.
>
> We will elaborate our discussion of the differences between the algorithms.  Even though at a superficial level they appear very similar, they have very different behavior guarantees and are appropriate for different circumstances.

---

### Official Review · Reviewer_pKTX · 2022-04-24
**Interpolating betwen Non-Monotonic Beam Search and Monotonic Monobeam.**

**Confidence:** 4
**Overall Score:** Accept

**Review:**

# Summary

Beam Search is a common search algorithm for limited memory. An unexpected property of beam search is that with larger beams, the solution quality can deteriorate. Prior work introduced a variant `monobeam` of beam search which guaranteed that for increased beam widths, the solution quality can only improve. This comes at the cost of a general worse solution quality.

In this work, the authors, present two techniques which interpolate between beam search and monobeam based on a parameter `n`. By enforcing that a subset of the slots in the beam exhibit monotonic behavior. The experiments show that indeed their algorithm interpolate between the extreme cases.


# Feedback

The paper is an incremental improvement to interpolate between two versions of beam search. It is easy to read, but some things need to be better expressed. First, the caption of a figure should contain all information for an informed reader to understand the figure. It is bad if a reader needs to search in the full text how to understand the figure. Currently, Figure 4 and 5 are very hard to grasp. The current content of the figure caption is redundant, as I can easily extract this information from the x- and y-axis, but I would need to know:
- for which algorithm is the figure
- over which things are the averages computed? (I assume same width, same `n` value, and different tasks)
- Which data point corresponds to which width? (Is left to right always increasing the width?)
- It seems the legend of each subplot is sorted by solution quality. Is this the case? By which metric are they sorted?
The same holds also for tables. For table 1, the rank correlation is computed based on which features? I neither understand the meaning of line 9 in Algorithm 1. Can you explain this to me again?
Regarding the experiments. If `n` is greater than beam width your algorithm collapses to beam search. It seems this also happens in Figure 4 and 5. For w=30, each subplot shows 3 different data points (monobeam, w=n/2, and all others are equal to beam), for w=100 it shows 4 and so one. Once the configurations do not collapse anymore, we see the interpolations. I would remove the collapsed configurations from the plots.

I would have expected more references. On a first glance, you might add
- citations for heuristic search
- beam search
- is there someone who first showed the non-monotonicity of beam search (maybe it is also in the original beam search paper, otherwise figure 1 from Lemons et al. 2022 shows this too)
Furthermore, you should add a citation to indicate where figure 1 comes from.

After seeing figure 1, my first thought was, I wondered what is the advantage of the monotonicity guarantee, if the expected quality is worse than the expected quality of beam search. Then I wanted to see some average results which you gladly provided in Figure 4 and 5. There we see that beam search is on average similarly well or even better than the algorithms with monotonicity guarantee. Which raises the question, why should I care for monotonicity. Can you please explain me why I should care for the monotonicity, if the non-monotonic algorithm is on average better?



## Minor Errors
- Introduction: "that are too large to solve with ..." -> Specify what you mean by too large.
- Move Algorithm 3 closer to the place where is is explained
- "and those new nodes may have incorrectly low cost-to-go estimates" -> Wrong. Should be: "the children might have incorrectly low cost-to-go estimates"

---

> ### Author Response · Authors · 2022-04-29
> **Author response**
>
> Many thanks for your helpful suggestions, which we will take into account in the camera-ready version if the paper is accepted to the workshop.
>
> Yes, the increasing beam widths do in these cases lead to increasing cpu time (left to right), although this is not necessarily always the case for beam search.  Yes, legends are sorted by average solution cost across all beam widths.
>
> Please note that Figures 4 and 5 show how the average cost varies as the beam is increased by large amounts (eg, roughly a factor of 3).  Nonmonotonicty is a property of beam search as the width changes on single instances, as shown in Figures 1, 2, and 3 for example. Averaging across instances obscures the phenomenon.  A user may be willing to sacrifice a small amount of solution cost (in expectation) in order to be able to more easily set the beam width.